# On the Role of Dietary Nitrate in the Maintenance of Systemic and Oral Health

**DOI:** 10.3390/dj10050084

**Published:** 2022-05-13

**Authors:** Ulrich Schlagenhauf

**Affiliations:** Division of Periodontology, Center for Oral Health, University Hospital Wüerzburg, 97070 Wüerzburg, Germany; schlagenha_u@ukw.de; Tel.: +49-931-201-72420

**Keywords:** nitric oxide, nitrite, nitrate, diet, oral, periodontitis, caries

## Abstract

The assessment of the significance of nitrates ingested with food has undergone a fundamental change in recent years after many controversial discussions. While for a long time, a diet as low in nitrates as possible was advocated on the basis of epidemiological data suggesting a cancer-promoting effect of nitrate-rich diets, more recent findings show that dietary nitrate, after its conversion to nitrite by nitrate-reducing bacteria of the oral microbiota, is an indispensable alternative source for the formation of nitric oxide (NO), which comprises a key element in the physiology of a variety of central body functions such as blood pressure control, defense against invading bacteria and maintenance of a eubiotic microbiota in the gut and oral cavity. This compact narrative review aims to present the evidence supported by clinical and in vitro studies on the ambivalent nature of dietary nitrates for general and oral health and to explain how the targeted adjuvant use of nitrate-rich diets could open new opportunities for a more cause-related control of caries and periodontal disease.

## 1. Introduction

Nitrate (NO_3_) is the oxidative product of nitric oxide (NO) and nitrite (NO_2_). As soil content, NO_3_ is an essential substrate for all plant growth and provides, among other things, the nitrogen required for the synthesis of nitrogenous amino acids. It is taken up by the plant roots and originates either from the microbial decomposition of organic waste or from the activity of certain soil bacteria utilizing atmospheric nitrogen for the synthesis of ammonia, nitrite and nitrate. Since available nitrate is the limiting factor for plant growth, it is deliberately added to the soil in agricultural crop production in the form of nitrate-containing fertilizers [1,2,3].

### 1.1. Significance of the NO-NO_2_-NO_3_ Cycle for Human Physiology

The metabolic NO-NO_2_-NO_3_ cycle is intimately associated with a multitude of essential physiological reactions within the human body. In 1998 Robert F. Furchgott, Louis J. Ignarro and Ferid Murad were awarded the Nobel prize in physiology for their “discoveries concerning nitric oxide (NO) as a signaling molecule in the cardiovascular system”. They were able to demonstrate that the central mechanism of physiological blood pressure control is the relaxation of vascular smooth muscle cells by the release of NO in the vascular endothelium, which originates from the oxidative conversion of the amino acid l-arginine into l-citrulline by a nitric oxide synthase [4]. In the human body, there are three different isoforms of nitric oxide synthase (NOS), all generating NO via the l-arginine to L-citrulline oxidation pathway [5]:Endothelial Nitric Oxide Synthase (eNOS)

eNOS is expressed by endothelial cells but has also been found in platelets, kidney epithelial cells, the placenta and neurons of the brain. Its activity is primarily regulated by a Ca^2+^-dependent binding to calmodulin; however, there are additional, Ca^2+^-independent pathways of eNOS activation, e.g., through fluid shear stress. Plasma concentrations of insulin or estrogen and vascular endothelial growth factor (VEGF) may influence the level of eNOS activation. Next to its dilating impact on all types of blood vessels, the eNOS-stimulated release of nitric oxide strongly inhibits platelet aggregation and adhesion, thus preventing any unwanted formation of thrombosis under physiological conditions. In addition, the release of nitric oxide triggered by eNOS reduces the synthesis of the chemo-attractive protein MCP-1 [6] and inhibits leukocyte adherence to vascular walls [7]. In patients with manifest vascular disease and in many individuals displaying only typical risk factors for developing cardiovascular disease, eNOS-mediated formation of nitric oxide in the endothelium is reduced. This is mostly accompanied by an elevated NADPH oxidase-dependent release of reactive oxygen species, which triggers an increased oxidative degradation of nitric oxide. In aged vessels, chronic oxidative stress may result in NOS uncoupling, i.e., the conversion of eNOS from a NO-synthesizing enzyme to an enzyme that generates oxygen radicals [8,9]. The results of a human intervention trial have shown that endothelial dysfunction may further originate from a lack of available l-arginine, as the supplementation of the diet with a daily dose of ~3 g l-arginine resulted in a significant reduction in manifest endothelial dysfunction in selected patient groups [10].

2.Neuronal Nitric Oxide Synthase (nNOS)

nNOS is constitutively expressed in central and peripheral neurons as well as in some other cell types. In the central nervous system (CNS), it mediates synaptic plasticity, including phenomena such as long-term potentiation or long-term inhibition, which are essential prerequisites for the ability to learn and the formation of memory. CNS-released nNOS may play an important role in the central regulation of blood pressure, as in an animal model, the blockade of nNOS synthesis in the hypothalamus led to systemic hypertension [11]. In the peripheral nervous system, nNOS is known to be a central regulator of smooth muscle tonus in intestinal peristalsis or penile erection and influences peripheral blood pressure through the activation of nitrergic nerves, which are exclusively stimulated by the release of nitric oxide. As with eNOS, the activity of nNOS is closely regulated by Ca^2+^-dependent binding to calmodulin.

3.Inducible Nitric Oxide Synthase (iNOS)

iNOS is expressed in a multitude of host defense and some other cell types in response to stimuli such as proinflammatory cytokines, bacterial lipopolysaccharides and some other molecules posing a threat to tissue integrity. Its activation induces a Ca^2+^-independent release of large amounts of NO aimed at destroying invaders such as bacteria or parasitic and tumor cells by exposing them to high levels of nitrosative stress. Unlike eNOS and nNOS, the activity of iNOS is not tightly controlled and counter-regulated. Excessive iNOS activation, therefore, often results in host tissue damage and is known to be an important aggravating factor in the etiology of periodontitis [12] and other chronic inflammatory conditions such as cardiovascular disease [13]. Large amounts of inflammation-derived NO entering the blood circulation have been identified as the main trigger for the development of septic shock [14].

### 1.2. Nitric Oxide (NO)

Due to its unpaired electron, nitric oxide is a very reactive signaling molecule acting via a variety of signaling modes. Most of them are based on the binding of NO to the haem group of soluble guanyl cyclase. This increases the intracellular concentration of the second messenger cyclic guanosine mono phosphate (cGMP), which is involved in a multitude of different secondary effects such as vasodilation, nerve signaling, mitochondrial biogenesis or angiogenesis [15]. NO is a very short-lived molecule with a half-life of only milliseconds. Under physiological conditions, it is rapidly oxidized to nitrate by oxyhemoglobin or to nitrite by the multi-copper oxidase ceruloplasmin [16]. Normal plasma levels of nitrate vary between 20–40 µM, those of nitrite between 50–300 nM and are stabilized by renal excretion; however, under severe systemic inflammatory conditions, plasma nitrite and nitrate levels may significantly exceed physiological levels due to a massive induction of iNOS activity. In cardiovascular diseases characterized by endothelial dysfunction and impaired eNOS activity, plasma concentrations of nitrate and nitrite, by contrast, are usually reduced [17].

## 2. NOS-Independent NO Generation from Dietary Nitrate

For many years the NOS-mediated oxidation of l-arginine was supposed to be the only source of NO generation in human physiology. The resulting plasma levels of nitrate and nitrite were considered merely oxidation or waste products of NOS-mediated NO-synthesis without any relevant beneficial physiological importance of their own. Only in the last few decades has a steady number of investigations revealed that NO-availability within the human body is not exclusively NOS-mediated, but is complemented, particularly under acidic and hypoxic conditions, by an alternative pathway of NO formation based on the reduction in inorganic nitrate and nitrite taken up with the diet [16]. Ingested dietary nitrate rapidly enters the bloodstream—75% of it is subsequently excreted by the kidneys, while the remaining 25% accumulates in the salivary glands in concentrations 20 to 100 times higher than plasma levels [15]. Commensal bacteria featuring nitrate reductases such as certain *Veillonella* or *Rothia* species that colonize the dorsum of the tongue, in turn, use salivary nitrate as an energy source by reducing it to nitrite. Since human cells themselves are not capable of this metabolic pathway, it is most likely a sign of a symbiotic relationship, as has been suggested by Lundberg et al. [16]. The resulting salivary nitrite is swallowed and, under the influence of gastric acid, rapidly protonates to nitrous acid (HNO_2_), which subsequently dissolves into NO and other nitrogen oxides [18]. NO formation dominates in the simultaneous presence of antioxidants such as ascorbic acid or polyphenols, which are abundant, e.g., in fresh vegetables and fruits. Exposure of the gastric mucosa to nitrite-rich saliva induces the formation of NO, which subsequently increases blood flow and the thickness of the mucous barriers [16]. In addition, the antibacterial efficacy of the stomach fluid is significantly enhanced by an increase in its nitrite content [18,19]. *Helicobacter pylori*, a pathogen closely associated with the development of gastric ulcers and stomach cancer and well adapted to survive under the very acidic conditions of the stomach, was readily eliminated in an in vitro study by the addition of 1 mM nitrite to the acidified culture media [20]. Intubation of critically ill blocks the flow of salivary nitrite to the stomach and thus intragastric NO formation. This has been suggested to be an important factor for an impaired survival prognosis of intubated patients due to the promotion of pathogen overgrowth and bacterial translocation through weakened epithelial barriers [21]. Furthermore, a diet-mediated increase in salivary nitrite and the acidic conditions of the stomach stimulate the formation of S-nitrosothiols, which are organic compounds or functional groups, displaying a nitroso group bound to the sulfur atom of a thiol [22]. They are known to be key components of physiological blood pressure control and potential drugs for the therapy of hypertension [23]. Red blood cells contain substantial amounts of S-nitrosohemoglobin, which are released under hypoxic conditions and stimulate vasodilation [24]. Salivary nitrite that has not reacted with other compounds during its passage through the stomach is eventually absorbed into the bloodstream as inorganic nitrite. This provides the body with an essential alternative source of NO, particularly under hypoxic conditions, when oxygen-dependent nitric oxide synthases may no longer be functional. Increased plasma levels of nitrite have been shown to exert a tissue-protective effect in ischaemia and to alleviate reperfusion damage in conditions such as myocardial infarction [16,25]. A transient increase in plasma concentrations of nitrate and nitrite, counteracting a hypoxia-related rise in oxidative stress, has been observed in expert breath-hold divers during deep dives [26]. The various pathways of the NO-NO_2_-NO_3_ cycle are schematically depicted in Figure 1.

## 3. Dietary Nitrate and the Maintenance of Oral Health

Although the role of the oral microbiota as a key element in the alternative formation of NO_2_ and NO from nitrate-rich foods has been known for many years, it is surprisingly only in recent years that the therapeutic and preventive prospects of a nitrate-rich diet have attracted the interest of dental researchers.

### 3.1. Impact of Dietary Nitrate on Caries Development

As already explained, salivary nitrite has a pronounced antibacterial effect that increases with decreasing pH values and easily penetrates microbial biofilms [19]. Increasing the level of nitrite in saliva by eating a diet rich in nitrates has therefore been proposed as a promising alternative strategy to control tooth decay. Scoffield et al. [27] evaluated the impact of nitrite in the culture media of *Streptococcus mutans* co-cultivated with *Streptococcus parasanguinis*. They observed a significant reduction in *S. mutans* growth rate and biofilm formation compared to the controls not being exposed to nitrite. In a subsequent animal study, inoculation of rats with *S. parasanguinis* prior to infection with *S. mutans*, supplemented with the addition of nitrite to the drinking water, resulted in a significantly lower number of newly formed carious lesions in enamel and dentin compared to the findings in the control animals, which did not consume nitrite-supplemented drinking water. These observations are supported by the results of Hohensinn et al. [28], who reported a significant increase in salivary NO_3_, NO_2_ and NO levels accompanied by a significant increase in mean salivary pH from pH 7.0 to pH 7.5 in healthy volunteers who regularly consumed a nitrate-rich beetroot juice.

Similarly, Burleigh et al. [29] observed a significant attenuation of salivary pH lowering in male endurance runners following the consumption of carbohydrate-rich snacks by the simultaneous ingestion of nitrate-rich beetroot juice.

More detailed insights into the role of salivary nitrate in promoting the establishment of a eubiotic non-cariogenic oral microbiome were recently provided by Huffines et al. [30]. They analyzed the impact of the addition of nitrite to the culture medium in a mixed in vitro biofilm model containing the commensal *S. parasanguinis*, and the cariogenic pathogens *S. mutans* and *Candida albicans. S. parasanguinis* showed a high degree of intrinsic resistance to the experimental increase in nitrosative stress and even increased it by the formation of peroxide, which in the presence of nitrite is readily converted to peroxynitrite a potent antibacterial compound. *S. mutans* and *C. albicans*, by contrast, were very sensitive to nitrosative stress and showed a significant reduction in growth rate and biofilm formation.

However, there are currently no data from controlled clinical trials to confirm these promising findings on the benefits of a nitrate-rich diet in controlling caries development.

### 3.2. Impact of Dietary Nitrate on the Development of Gingivitis and Periodontitis

For many years the focus of research efforts in periodontology was mostly on the role of iNOS activation in periodontal tissue breakdown and the development of therapeutic approaches to control it [31,32]. As described before, salivary nitrite concentration may either reflect the metabolic activity of nitrate-reducing oral bacteria and/or are influenced by a local efflux of iNOS-generated NO and NO_2_ from inflamed tissue sites in gingival and periodontal inflammation; therefore, several authors assessed the usefulness of NO and NO_2_ level measurement in gingival crevicular fluid and saliva as diagnostic markers of periodontal health with diverging results. Topcu et al. [33] investigated nitrate and nitrite levels in saliva and gingival fluid and reported significantly increased nitrite levels in the gingival fluid of gingivitis patients compared to healthy controls. In an in vitro study, Hussain et al. [34] exposed cultured oral keratocytes to the culture supernatants of a variety of oral microorganisms, including *Aggregatibacter actinomycetemcomitas*, *Campylobacter rectus*, *Porphyromonas gingivalis, Streptococcus salivarius* and *C. albicans*. They observed significantly different concentrations of nitrite released into the cell culture medium by the keratocytes in response to the exposure to the different bacterial supernatants, with the supernatants of *A. actinomycetemcomitans* and *C. rectus* eliciting the strongest response. The same group of researchers demonstrated in a later study [35] that the number of iNOS and adrenomedullin (AM) expressing cells was three-fold higher in inflamed than in healthy gingival tissues. They also observed that salivary NO and AM levels were significantly higher in patients with aggressive periodontitis than in patients with chronic periodontitis or gingivitis. The mean NO levels in gingival crevicular fluid differed significantly between aggressive periodontitis and chronic periodontitis, as well as between chronic periodontitis and gingivitis. Han et al. [36] showed similar findings in a cohort of periodontitis-affected elderly Koreans. In contrast, Aurer et al. [37] reported decreased salivary levels of NO and NO_2_ in periodontitis patients.”

Again, only recently the significance of dietary NO_3_ consumption for the maintenance of periodontal health has been the subject of scientific investigations. In a controlled clinical intervention trial, Jockel-Schneider et al. [38] assessed the influence of the regular consumption of a nitrate-rich lettuce juice beverage on the manifestation of chronic gingivitis in a cohort of treated periodontitis patients regularly seeking periodontal aftercare. They were able to show that the targeted daily intake of 200 mg of dietary NO_3_ over an observation period of two weeks after professional mechanical plaque removal (PMPR) led to a halving of gingival inflammatory expression compared to baseline. By contrast, in controls, having consumed an identical lettuce juice but without any NO_3_ content, the ameliorative impact of PMPR on gingival inflammation was only small and not significant when compared to baseline levels. Microbial samples taken from the periodontal pockets of the study participants at the onset and at the end of the study were subjected to whole-genome sequencing at the genus level. In the test group consuming a nitrate-rich diet, a significant shift in the composition of the different bacterial genera was observed. At the end of the study, the frequency of bacterial genera associated with periodontal inflammation such as *Treponema* and *Prevotella* was significantly reduced, while the presence of genera associated with periodontal health such as *Rothia* and *Neisseria* was significantly increased. In control group patients, by contrast, despite being equally subjected to PMPR but having consumed a nitrate-depleted diet, differences in the composition of the microbiota between baseline and reevaluation were only small and not significant [39]. The persistence or rapid recurrence of bacterial dysbiosis in the control group clearly proved that the effect of mechanical plaque removal may only be symptomatic, while the significant resolution of bacterial dysbiosis observed in the test group suggests that dietary aspects may play an important causative role in the overgrowth of disease-promoting oral pathobionts. These data are partially supported by the findings of another clinical investigation. Woelber et al. [40] reported an inhibitory influence of a daily diet rich in omega-3 fatty acids, antioxidants and dietary NO_3_ on the manifestation of gingivitis in a cohort of individuals who had been habitually consuming a Western diet containing over 45% processed carbohydrates before the onset of the study; however, due to the combined consumption of various food components with anti-inflammatory properties evaluated in this trial, it is not possible to assess the individual significance and contribution of dietary NO_3_ consumption to the results. Nevertheless, despite these promising data on the efficacy of a high-nitrate diet as an adjuvant in preventing and treating periodontal inflammation, the available evidence must be considered preliminary until additional results from larger controlled clinical trials are available.

### 3.3. Impact of Oral Antiseptics on Salivary NO_2_ Levels

In many industrialized countries, the regular use of an antibacterial mouth rinse is a popular supplement to daily toothbrushing. This may be problematic as an undifferentiated long-term use of oral antiseptics not only inhibits the growths of pathogens but may concomitantly impair the vitality and metabolism of oral symbionts such as the aforementioned nitrate-reducing microbiota on the tongue. Kapil et al. [41] reported that the use of an antiseptic chlorhexidine-based mouthwash suppressed microbial nitrate reduction and resulted in a significant increase in systolic blood pressure of young healthy volunteers. Other authors have confirmed this finding [42], while in epidemiological surveys, the frequent use of mouthwashes was associated with elevated incidences of hypertension [43] and type 2 diabetes [44], as the availability of nitrate ingested with the daily diet has shown to be an important modulator of insulin resistance in obese adults [45] and metabolic syndrome patients [46]; therefore, although the use of proven oral antiseptics such as chlorhexidine is still considered safe in principle, current guidelines for the prevention and treatment of caries and periodontal disease recommend their use as an adjunct to mechanical plaque control only for certain indications and limited time spans [47,48].

## 4. Nitrate-Rich Vegetables

Nitrate-rich vegetables include many popular varieties such as lettuce, rocket, chard and spinach, cabbage, radish or beetroot (see Figure 2). The actual nitrate content of an individual plant does not only depend on the variety. It is decisively influenced by the amount of fertilization, seasonal and geographical parameters and the time of harvest within a day [49]. The latter reflects the fact that in photosynthesis, sunlight is not only the main source of energy for the reduction in carbon dioxide but also for the reduction in inorganic NO_3_ taken up from the soil [50]; therefore, NO_3_ accumulates in the plant during the night and is consumed again during the day by photosynthesis. This makes it very difficult or impossible for agricultural producers to predictably standardize the NO_3_ content of harvested vegetables within a narrow range. Furthermore, cooking or frying may noticeably change the initial nitrate content of raw, unprocessed vegetables [51]. The amount of NO_3_ ingested with the consumption of unprocessed vegetables can therefore only be roughly estimated with the help of comparison tables [49]. A more accurate monitoring of NO_3_ intake requires either the consumption of processed vegetable products with a defined NO_3_ content, as in the study by Jockel-Schneider et al. [38], or the direct measurement of salivary NO_2_ levels, which increase after nitrate-rich meals due to the metabolic activity of the nitrate-reducing oral microbiota. This can most easily be achieved with specific, commercially available test strips, which have become popular among athletes, as a nitrate-rich diet can significantly increase endurance and high-intensity performance by improving coronary and muscular blood flow [52,53].

## 5. Diet and Oral Health—Historical Aspects

The significance of malnutrition and a nutrient-deficient diet for the development of bacterial dysbiosis and the onset of caries and periodontal disease is further supported by the findings of an unusual archaeological study. Its authors assessed human skulls from the end of the Neolithic period to the beginning of the Industrial Revolution for the health status of the dentition and were able to recover identifiable microbial DNA specimens from dental calculus adhering to the teeth [54]. Analysis by whole-genome sequencing allowed the identification of individual microbial species and the determination of microbial diversity. The results link the historical appearance of caries and periodontal disease as widespread human ailments to the change in lifestyle from hunting and gathering to farming. Hunter-gatherers usually consumed a very varied diet, which did not only provide all nutritional prerequisites for maintaining systemic health, but also the necessary growth substrates for a very diverse and health-compatible oral microbiota. By contrast, the transition to a sedentary, farming lifestyle at the end of the Neolithic was accompanied by a significant reduction in dietary diversity resulting in a significant reduction in oral bacterial diversity, which again favored the dysbiotic overgrowth of disease-promoting pathobionts and the onset of periodontal disease. A subsequent comparison of medieval calculus samples with those from the Industrial Revolution period revealed an additional reduction in microbial diversity and the ubiquitous presence of cariogenic bacteria by the now widespread consumption of refined sugar, which was associated with an enhanced deterioration of oral health. While not specifically assessed by the authors, the frequent consumption of non-cultivated edible wild plants by hunter-gatherers will most likely have provided them with significant amounts of dietary NO_3_, as a variety of edible wild plants naturally contain high levels of NO_3_ [55].

## 6. Controversies about the Impact of Dietary Nitrate on General Health

All findings presented so far in this review suggest a significant physiological role of dietary NO_3_ for microbial homeostasis and the maintenance of oral and systemic health; however, NO, NO_2_ and NO_3_ are molecules with an ambivalent nature. Although they are crucially important for the proper maintenance of physiological conditions, they may act as toxic agents at the same time when released by host defense cells under inflammatory conditions, as described before. In most countries, there are legal limits for NO_3_ and NO_2_ in drinking water. They were established in order to prevent the manifestation of methemoglobinemia in newborns, as fetal hemoglobin is particularly susceptible to oxidation by NO_2_ due to a transient deficiency of methemoglobin reductase in neonatal erythrocytes [55], which may result in the manifestation of cyanosis. While contaminated drinking water is a very unlikely cause of methemoglobinemia in industrialized countries, according to case reports [56], the feeding of infants with microbially tainted nitrate-rich foods may still pose a potential problem.

Another major concern regarding the safety of dietary NO_3_ raised by toxicological and epidemiological investigations is the formation of carcinogenic compounds stimulated by the ingestion of nitrate-rich foods or nitrate-rich drinking water [57]. This is based on the findings from various epidemiological studies that the frequent consumption of processed meat products such as ham, bacon, salami or hot dogs, which are usually cured by nitrite- and nitrate-containing salts, is associated with an increased risk of cancer [58]; however, this conclusion is still not generally consented by all toxicological experts and could not be validated definitively by recent meta-analyses [59]. As described before, ingested NO_2_ dissolves under the acidic conditions of the stomach to nitrosyl (NO_-_), a very reactive molecule, which may interact with biogenic amines from consumed meat to form carcinogenic nitrosamines. The clinical significance of this potential pathway of cancerogenesis however has been questioned by the fact that even in Australia, a country with one of the highest consumption rates of red meat in the world, only about 1% of the total uptake of nitrate originates from the consumption of processed meat, compared to more than 80% from the intake of vegetables [60]. The consumption frequency of nitrate-rich green leafy vegetables, however, is known to be even inversely correlated with the risk of developing cancer [61,62]; therefore, while the impact of NO_2_ and NO_3_ in processed meat products on cancer formation is still subject to an ongoing debate, the regular consumption of fresh nitrate-rich vegetables may be considered safe. According to a European Food Safety Authority (EFSA) expert panel on contaminants in the food chain, daily consumption of more than 400 g of mixed vegetables will not exceed the so-called Acceptable Daily Intake (ADI) of 222 mg of NO_3_, set by the WHO/FAO for a 60 kg person [49]. It should be noted that the ADI does not represent a toxicity or danger threshold but rather corresponds to the amount that can be consumed daily over a lifetime without significant health risks.

## 7. Dietary Nitrates—Implications for Clinical Practice

Despite the available evidence for the importance of dietary deficits as significant risk factors for the deterioration of systemic and oral health, current evidence-based guidelines for the treatment of periodontal disease do not assign a prominent role to targeted nutritional counseling. This may be because there is still insufficient evidence of the efficacy of this approach based on the results of long-term controlled clinical trials [63], whereas there is ample evidence of the efficacy of established non-cause-oriented strategies based on mechanical plaque control in reducing disease progression to clinically acceptable levels in most patients [48]. Furthermore, it may often be very difficult to permanently correct a patient’s disease-promoting dietary habits, as suggested by the disappointing results of a systematic review on the long-term compliance of hypertension patients with therapeutic dietary instructions [64]. Nevertheless, at least in cases of rampant caries or severe periodontitis in patients who are simultaneously affected by chronic systemic inflammation, a purely mechanical strategy based on plaque control is unlikely to be sufficient to adequately control disease activity and will benefit from the adjuvant identification and correction of disease-promoting nutritional deficiencies.

## 8. Conclusions

Dietary nitrate is an indispensable, alternative source for the formation of nitric oxide, which is a key signaling molecule involved in a multitude of essential bodily functions, including the maintenance of a disease-preventive balance between the human microbiota and its host. The available evidence from controlled clinical trials suggests that the adjuvant consumption of a varied nitrate-rich diet may have the potential to become a useful supplement to established therapeutic and preventive measures for the recovery and stabilization of oral health and oral bacterial eubiosis.

## Figures and Tables

**Figure 1 dentistry-10-00084-f001:**
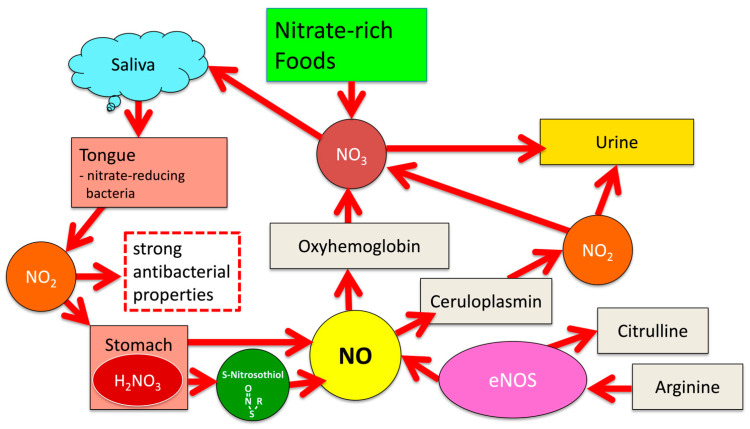
Pathways of the NO-NO_2_-NO_3_ cycle.

**Figure 2 dentistry-10-00084-f002:**
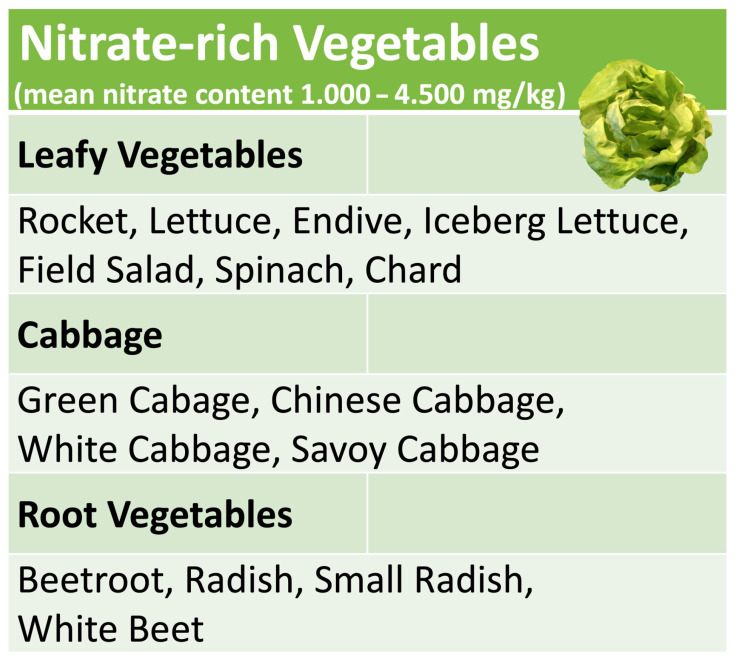
Common nitrate-rich vegetables.

## Data Availability

Not applicable.

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
