# Peer review of "On the Role of Dietary Nitrate in the Maintenance of Systemic and Oral Health"

_dentistry, 2022, doi:10.3390/dj10050084_

Round 1
Reviewer 1 Report
Dear author,
This is an interesting narrative review about the effect of dietary nitrate on oral health, mainly dental caries and periodontal diseases. The queries are provided below:
Abstract
- The abstract must not be seen as an Introduction section. Author is encouraged to summary the main findings of the present study in this section.
- The aim of the present study report in this section is too narrow. As the present study evaluated several aspects of the nitrate in oral health, the aim must be rewritten.
Introduction
- This section is well-written and the information provide are very important to understand the next sections of the study.
Impact of dietary nitrate on the development of gingivitis and periodontitis
- Author report inconclusive results for studies #31-33. However, it is not clear what “inconclusive” means. Please, be more specific.
- Based on the available evidence, nitrate diet was used as an adjuvant periodontal therapy. Include this information in the last sentence of this section.
Impact of oral antiseptics on salivary NO2 levels
- The clinical impact of the Kapil’s study (reference #41) must be considered when interpreting their results. In addition, regular use of chlorhexidine is not recommended.
- I was not able to understand the last sentence of this section. The use of mouthwash is very safe and no current clinical study was able to demonstrate systemic impairments in individuals using mouthwash. What is the harm of a patient with diabetes type 2 when using a mouthwash? This is not clear.
Controversies about the impact of dietary nitrate on general health
- Try to mitigate the first sentence of this section. Based on the available evidence, it is not possible to provide such a strong statement.
- Other evidences are available providing further discussion (and controversies) on this topic:
- Hosseini et al. Nitrate-nitrite exposure through drinking water and diet and risk of colorectal cancer: A systematic review and meta-analysis of observational studies. Clin Nutr. 2021 May;40(5):3073-3081. doi: 10.1016/j.clnu.2020.11.010. Epub 2020 Nov 28.
- San Juan, A. F., Dominguez, R., Lago-Rodríguez, Á., Montoya, J. J., Tan, R., & Bailey, S. J. (2020). Effects of dietary nitrate supplementation on weightlifting exercise performance in healthy adults: A systematic review. Nutrients, 12(8), 2227.
- Remington, J., & Winters, K. (2019). Effectiveness of dietary inorganic nitrate for lowering blood pressure in hypertensive adults: a systematic review. JBI Evidence Synthesis, 17(3), 365-389.
Dietary nitrates - implications for clinical practice
- Again, the available evidence about this topic is, indeed, interesting and worth of further investigations, but it is not strong. Mitigate the first sentence.
- In fact, the current guideline for the treatment of periodontitis stages I-III was not able to provide a solid recommendation regarding nutritional counselling and the treatment of periodontitis (Sanz et al., 2020 - Treatment of stage I–III periodontitis—The EFP S3 level clinical practice guideline).
- The following sentence needs citation: “In these severe cases current guidelines advocate the adjuvant use of chemotherapeutic agents, whose negative impact on systemic health due to the unavoidable concomitant killing of beneficial, symbiotic bacteria has been described before.” This is very important, as the previously mentioned guideline do not advocate for the use of adjuvant substance in the treatment of periodontitis.
Conclusion
- The last sentence must also be mitigated, as there is no high-quality available evidence to support this statement.
Reviewer 2 Report
The manuscript is fairly easy to read and comprehend. Some English errors in the Concusion section. Otherwise, this reviewer is generally happy about the content.
Author Response
Thank you very much for your review. I have revised the Conclusion section and tried to improve the English language. Please find the revised manuscript with highlighted changes suggested by you and the other reviewers in the attachment.

Reviewer 3 Report
The manuscript is a narrative review. Hence, there are limitations.
I recommend authors go for a systematic search of literature and incorporate the literature accordingly.
Reorganisation of the content is required
This can be discussed as animal studies, microbiological on culture media and biofilms, invitro studies on human specimens, clinical studies on humans and trials.
Have found some literature which was not included or discussed in the manuscript:
Nitrate-rich beetroot juice offsets salivary acidity following carbohydrate ingestion before and after endurance exercise in healthy male runners.
Nitrite Triggers Reprogramming of the Oral Polymicrobial Metabolome by a Commensal Streptococcus
Antimicrobial can be considered instead of Antiseptic
The role of cooking or processing on the availability of Nitrates needs to be discussed; Role of other foods on the absorption and availability of the same needs to be discussed.
Round 2
Reviewer 1 Report
Congratulations for the hard work and the improvements. Overall, the quality of the manuscript increased significantly. However, in one sentence, a citation is missing as stated below.
Dietary nitrates - implications for clinical practice
- Provide citations for the following sentence: “Recent systematic reviews of periodontal preventive strategies based on professional plaque removal and the improvement of personal oral hygiene clearly proved, that this approach, while not cause-related, may be still a clinically successful and feasible intervention to offset the negative impact of dietary deficits on oral health in most patients.”
Reviewer 2 Report
The manuscript gave a brief account of the source of NO that is required in the NO pathway, without much description of the pathway itself. The authors then continued to describe the potential impact of NO on oral health (viz. caries and periodontitis) and systemic health (mostly on blood pressure control). With the limited description (only on bold pressure and minimally on effect upon Helicobacter pylori) on the general systemic health, the title of the manuscript is too broad and over-representing its contents. Other than that, the manuscript is quite easy to read and follow.
There is, however, some redundancy in the writing. Example in Introduction [line 24]: ... Is an essential broth substrate for all plant growth and provide, among other things, for nitrogen required for the synthesis of nitrogenous amino acids.
Some other comments are listed below.
There are far too many repetitive or unnecessary conjunctions and/or adverbs. Examples [line 42-45]: "also" appeared 3 times; "like" followed by "for example"; "as well as" can be replaced by 'and' there.
Spelling mistakes that affect meaning of the writing: "Th results..." [line 54] / "Due to is unpaired electron..." [line 80]
There are inconsistencies when stating the name of the bacterial species. It is good to see that the full name of a bacterium was given upon first appearance in the text. However, abbreviating the genus name was done haphazardly – sometimes, it was abbreviated (e.g. line 153-155), whereas full names were given all the times in certain paragraph (e.g. line 184-188).
Line [188-194] may be rewritten as follow to aid readability: "The same group of researchers reported in a later study (35), that the number of iNOS and adrenomedullin (AM) expressing cells was three-fold higher in inflamed than in healthy gingival tissues. They also reported that salivary NO and AM levels were significantly higher in patients with aggressive periodontitis than in patients with chronic periodontitis or gingivitis. The mean NO levels in gingival crevicular fluid differed significantly between aggressive periodontitis and chronic periodontitis, as well as between chronic periodontitis and gingivitis. Han et al. (36) reported similar findings in a cohort of periodontitis-affected elderly Koreans. In contrast, Aurer et al. (37) reported decreased salivary levels of NO and NO2 in periodontitis patients."
Also, for the conclusion: "...alternative source for the formation of nitric oxide, WHICH IS a key signalling molecule involved in a multitude..." [line 340-341]; and to delete the word 'first' in the second sentence.
Reviewer 3 Report
All my comments are addressed
